# Assessment of Hydrobiological and Soil Characteristics of Non-Fertilized, Earthen Fish Ponds in Sindh (Pakistan), Supplied with Seawater from Tidal Creeks

Asma Fatima [1], Ghulam Abbas [1] and Robert Kasprzak [2,*]

1 Centre of Excellence in Marine Biology, University of Karachi, Karachi 75270, Pakistan;
asmafatima516@gmail.com (A.F.); ghulamabbas@uok.edu.pk (G.A.)
2 Department of Ichthyology and Biotechnology in Aquaculture, Institute of Animal Sciences,
Warsaw University of Life Sciences—SGGW, 02-787 Warsaw, Poland
* Correspondence: robert_kasprzak@sggw.edu.pl

**Abstract:** In this study, the suitability of four earthen, seawater ponds located in the Thatta district of Sindh province (Pakistan) was evaluated for the purpose of semi-intensive mariculture, which remains to be a severely underdeveloped branch of the agricultural industry of this populous Asian country. Initial pond soil probes were promising, as they showed a high clay and silt content. Monthly water samples were obtained in the year 2019 (from January to December), which allowed for the monitoring of water parameters, as well as the identification and relative quantification of planktic populations. As a result, the monthly variations of basic water parameters were found within optimal ranges for planktic growth (water temperature, salinity, pH, transparency, and dissolved oxygen). Bacillariophyta was the largest phytoplanktic group, with the most dominant species being *Sundstroemia setigera*, followed by the cyanobacteria *Oscillatoria limosa*. Copepoda was the most numerous group of identified zooplankton, followed by tintinnids and foraminiferans. Total suspended solids (TSS) calculations indicated up to nine-fold month-to-month reductions of planktic biomass, observed in the form of diminishing Bacillariophyta (December) and Copepoda (June and December). In conclusion, the studied ponds appear to be suitable for semi-intensive mariculture activity due to the abundance of diverse planktic forms (mainly Copepoda—preferable natural food for commercially important fish species), which was achieved even without the use of fertilizers. However, significant drops of planktic biomass may still occur, which implies the need for regular water monitoring procedures, which would in turn allow fish producers to implement periodical adjustments to the administered feeding rates with artificial diets.

**Keywords:** planktic biomass; phytoplankton; zooplankton; water parameters; semi-intensive aquaculture; mariculture

## 1. Introduction

Plankton constitute an important role in the trophic chain, establishing the energy flow between primary producers and planktivorous fish and shellfish species [1,2]. Therefore, plankton have been deemed invaluable for a sustainable pond-based aquaculture, as the presence of abundant pond biota is necessary to fulfill the nutritional requirements of many farmed fish [3,4] and shrimp species [5,6]. Worldwide, numerous studies focused on planktic populations are being conducted in aquaculture to assess pond water quality, productivity, and other characteristics or interactions [7–9].

Large amounts of nutrients stored in soil may stimulate the growth of plankton in earthen ponds, resulting in increasing final pond productivity [10]. Soil nutrient characterization is governed by soil texture, categorized into sand, silt, and clay [11], implying that not every location is suitable for the construction of inland aquaculture ponds.

Estimating the water quality of ponds is necessary for determining the potential courses of limnological change, and the measurements of physicochemical parameters (such as temperature, pH, total hardness, alkalinity, potassium, phosphate, nitrate, sulphate, dissolved oxygen—DO, and biological oxygen demand—BOD) indicate the suitability of the tested water for various forms of aquatic life [12,13]. Furthermore, water quality can directly affect different behavioral and physiological actions of fish or crustaceans, such as feeding, breeding, swimming, metabolism, and excretion [14]. Therefore, good water quality is indispensable for better growth, survival, and higher production of cultured fish or shellfish [15].

In semi-intensive earthen ponds, the most conventional systems of aquaculture involve both the promotion of natural food sources using fertilizers and the administration of artificial diets designed directly for the farmed species [16,17]. Due to such practices, the abundance of planktic communities grows to a certain level with the concurrent rising concentration of nitrogen, phosphorus, and other biogenic compounds [18,19]. However, it is presumed that the quality of pond water deteriorates significantly by providing a surplus of commercial feeds and (especially) fertilizers, resulting in a low concentration of DO and excessive accumulation of $NH_3$, $NO_2^-$, and phosphates [20,21]. In addition, the excess of nutrients causes phytoplanktic blooms in ponds, further intensifying the anoxic conditions and increasing water pollution [22–24]. Apart from the overabundance of nutrients, temperature fluctuations may also suppress the primary productivity of water [25,26].

The management of coastal saltwater ponds is different from their freshwater counterparts, as fertilizers do not usually need to be used prior to stocking. Meanwhile, the constant flow of seawater stemming from sources such as tidal creeks, stable pH and temperature, low salinity fluctuations, and high evaporation rates all contribute to preventing methane production in the sediments [27]. However, similarly as in freshwater pond systems, their aging directly results in decreasing growth rates of farmed fish [28].

The mariculture industry has generally flourished worldwide over the last couple of decades due to an ever-growing demand for aquatic products, occurring alongside the rapid increase in global population [29]. Meanwhile, mariculture in Pakistan is still in its experimental stages; currently, there is only minor shrimp production [30,31], and coastal finfish culture technology has not been successfully developed yet for commercial purposes, especially when compared to other relevant countries [32,33]. Only but a handful of culture-oriented studies have been done locally on marine fish, including the goldsilk sea bream, *Acanthopagrus berda* [34–36], and the mangrove red snapper, *Lutjanus argentimaculatus* [37–39], but none of these works included any form of suitability assessment of coastal brackish ponds. A few rudimentary analyses were only performed for inland ponds [40–42].

Therefore, to obtain baseline data, we executed a preliminary trial with three marine fish species stocked in polyculture (a first such trial ever attempted in Pakistan) in non-fertilized coastal ponds, with water supplied from tidal creeks. Over the whole calendar year of 2019, we evaluated the abundance and diversity of planktic populations in the ponds, along with the assessment of physicochemical parameters of soil and water. The aim of this work was to study the major outlines of the limnological change processes occurring in such non-fertilized ponds, especially with regard to the sustainability of valuable zooplanktic communities within these water bodies. Such knowledge may be later incorporated into advanced fish production protocols, designed specifically for semi-intensive ponds located in this area. In the long term, such efforts should prove beneficial for the heavily underdeveloped Pakistani mariculture industry.

## 2. Materials and Methods

### 2.1. Experimental Site and Pond Management

Garho Fish Farm of Sindh Fisheries Department (24°19′58.1″ N, 67°34′51.4″ E), located near Mirpur Sakro, Thatta district, Sindh, Pakistan, was chosen for the study. A total of four semi-intensively managed fish ponds, which were built 10 years prior (but were only

tentatively used for fish and shrimp farming activities), were evaluated. Each rectangular pond covered the area of ~1.43 ha (130 × 110 m) and had a depth of 0.9–1.5 m (approximate volume of water was 15,000–20,000 m$^3$). The dykes, monk, and canals were all freshly repaired (the ponds were out of use at the time, and lime was used at 1–2 t ha$^{-1}$ as a disinfectant before stocking) [43,44]. Seawater from the Garho channel (originating from tidal creeks) was directly pumped into the ponds from the canals (where the water was stored initially), with a minor but constant influx of new saltwater from the channel throughout the entire study, mitigating the evaporation. No additional chemical fertilizers were used during the study period. All ponds were stocked with commercially important marine fish species (2000 fish for each): goldsilk seabream *A. berda* (mean weight = 17.4 g, mean total length = 112.5 mm), yellowfin seabream *Acanthopagrus latus* (18.6 g, 119.7 mm), and milkfish *Chanos chanos* (25.9 g, 138.2 mm). Supplementary feed (32.5% protein and 12.5% lipids) was administered daily at approximately 2.5% total fish body mass per day. The total duration of the study spanned one year, from January (I) to December (XII) 2019. Afterwards, water was drained from the ponds and all fish were gathered to evaluate gross pond production, which turned out to be 6053 ± 358 kg ha$^{-1}$ year$^{-1}$.

### 2.2. Soil Sample Collection and Analyses

Soil samples were collected from ponds before fish stocking. They were obtained from the upper 5 cm layer using a standard, handheld bottom core sampler (5 cm diameter) and were collected from 10 different, evenly spread out sites of each pond, and combined into one sample pond. Such a sampling procedure is recommended for aquaculture ponds [45]. Afterwards, the samples were analyzed in an external laboratory using methods commonly applied to soil [46]. In detail, the performed analyses included the estimation of gross texture (percentages of clay, silt, and sand), chemical parameters (pH, base unsaturation, cation-exchange capacity, exchangeable acidity), biochemical composition (total: carbon, nitrogen, sulfur, and phosphorus, as well as dilute-acid-extractable phosphorus and carbonates), and mineral content (Ca, K, Na, Mg, Fe, Al, Mn, B, Ba, Zn, Cu, Co, Mb, Pb).

### 2.3. Water Parameter Measurements and Calculation of Total Suspended Solids

The following physicochemical properties of pond water were investigated monthly during 9:00–11:00 AM for a period of one year: temperature with digital thermometer; salinity with digital refractometer (S/Mill-E, ATAGO, Tokyo, Japan); pH with digital pH-meter (EzDO 6011, GOnDO Electronic, Taipei, Taiwan); transparency of water with Secchi disk; DO, ammonia, nitrates, and phosphates were monitored with portable test kits (Merck KGaA, Darmstadt, Germany); potassium, calcium, total hardness, and alkalinity were measured with a water quality device (TitraLab AT1000, Hach Company, Loveland, CO, USA) and analyzed accordingly [45].

Total dissolved solids (TDS) were measured with a digital TDS meter (CD 610, Milwaukee Instruments, Rocky Mount, NC, USA), while total solids (TS) was calculated using the following equation: TS = Dry mass of sample/Volume of sample (ten 200 mL samples/pond, per month). Finally, the planktic biomass was estimated through the calculation of total suspended solids (TSS), which was performed using the equation TSS = TS − TDS [47].

### 2.4. Plankton Sample Collection and Analyses

Plankton samples from five selected sites of each pond were taken monthly from January to December 2019 (at 09:00–11:00 AM) by using a conical plankton net (56 μm mesh size). Given the fish culture context of the study, zooplankton was the primary focus of sampling; therefore, the used mesh size was higher than in nets designed specifically for the pickup of phytoplankton (20–30 μm), but it was a justifiable compromise as it was already more than 2x smaller than the suggested mesh size for tropical zooplankton [48]. Each time, the net was hauled horizontally through the water column and was dragged in the water for approx. 15 m, about 5 cm below the water surface. The net was then lowered close to the bottom and then slowly lifted to allow water to be filtered. The net was rigged with



additional weight to enhance vertical sinking. Wet samples (30 mL) were then obtained by using the cod end attached to the bottom of the net and were immediately fixed and preserved in plastic bottles containing 5% formalin solution, and were taken for investigation using a conventional upright light microscope (M11, Wild Heerbrugg, Heerbrugg, Switzerland), through a Sedgewick-Rafter counting chamber. The $100\times$ magnification objective (oil-immersion) was used to assess the relative counts of phytoplankton [49], while the 4x-40$\times$ objectives were used for zooplankton [50]. Briefly, 6 mL were picked up from each vigorously stirred, fixed sample and were analyzed in six 1 mL sub-samples for improved relative quantity estimation of the identified cells/organisms [51].Each time after pouring 1 mL of subsample onto it, the counting chamber was placed on the stage of the microscope and it was allowed to settle for 15–20 min. Plankton counting was done from one corner to the other, with the chamber being moved horizontally along each row of squares, and all organisms in each square of every row were counted. Relative quantity was indicated as a percentage of the counting chamber's squares in which each identified planktic species/taxon was found during the monthly microscopic observations of all obtained samples. This estimation was translated into a five level scale, from "−" (absent, 0%), "+" (less common, 1–20%), "++" (common, 21–40%), "+++" (abundant, 41–70%), to "++++" (most abundant, 71–100%). If no particular specimen was encountered, it was recorded as a zero count. Most of the time, a tally of approximately 150–300 specimens per row resulted in this species/taxon being considered as "most abundant". Plankton was identified categorically by using key guides and relevant literature [52,53].

*2.5. Statistical Analyses*

Pearson correlation coefficient was calculated to determine the correlations between soil and hydrological parameters through SPSS software (Version 16.0). Differences in monthly TSS measurements were analyzed using Duncan's new multiple test range, and are presented as mean with the standard deviation ($\pm$SD).

## 3. Results

*3.1. Soil and Water Parameters*

The results of soil sample analyses are presented in Table 1, and correlation coefficients between soil variables are presented in Table 2. A highly significant ($p < 0.01$) negative correlation was found between calcium and total nitrogen content, whereas significant ($p < 0.05$) correlations were found between organic carbon and calcium (positive), phosphorus (negative), and total nitrogen (negative).

The combined results of hydrological parameters are presented in Table 3, while correlation coefficients between chosen variables are presented in Table 4. Water temperature exhibited significant correlation ($p < 0.01$) with the pH, nitrates, transparency, and phosphates, while there was negative correlation with the DO. Phosphates showed significant ($p < 0.05$) negative correlation with nitrates, potassium, and transparency, whereas there was positive correlation with DO. The DO positively correlated ($p < 0.01$) with the pH and nitrates but showed negative correlation with transparency. Finally, pH also negatively correlated ($p < 0.01$) with transparency.

TSS was calculated separately for each pond and the results are presented in Table 5. Statistical analysis revealed that statistically significant ($p < 0.01$) drops of TSS were found in June and November–December.

*3.2. Phytoplankton Quantification and Identification in Fixed Sedimentary Samples*

A total of 64 planktic species were identified from the studied ponds and were arranged into 19 clades/categories. Of these, 27 species of phytoplankton were grouped into 4 clades and are displayed in Table 6. Bacillariophyta were by far the largest group (20 species), with the most dominant species being *Sundstroemia setigera* (highest abundance found in the month of II–IV, X–XI), followed by *Skeletonema costatum* and *Chaetoceros atlanticus* (both peaked in XI). *Oscillatoria limosa* (Cyanophyta) reached similar abundance in IV–X (with

drops in VI and IX). *R. setigera, Bacillaria paxillifera* and *O. limosa* were found throughout the entire year (within their different life stages), while *Gyrosigma spencerii* was found in all months except VII. Meanwhile, Dinoflagellata and Prymnesiophyta were identified only occasionally—only *Tripos furca* was found in a span of three months (X–XII).

### 3.3. Zooplankton Quantification and Identification in Fixed Sedimentary Samples

The zooplankton was arranged into 15 clades (a total of 37 taxa), as presented in Table 7. The most dominant group were Copepods, in which calanoids were the most abundant in IV and VII–IX months, cyclopoids in III and IX, and harpacticoids in III. Females carrying egg sacs were only sometimes distinguished among each of the three copepod orders, while copepod nauplii were classified as "most abundant" in II–IV months. Other than copepods, the most dominant occurrences of zooplankton were medusae (Scyphozoa) in I, *Tintinnopsis* spp. (Tintinnida) in X–XI, and various foraminiferans in X. Calanoids, cyclopoids, harpacticoids, copepod nauplii, *Limacina inflate, Doliolium denticulatum*, nematode worms, and unidentified eggs were observed during the whole study period, whereas scyphozoans were found in all months except VIII.

**Table 1.** Chemical properties of bottom soil of seawater ponds at Garho Fish Farm, Thatta, Pakistan.

| Variable | Mean | SD [1] | Min. [2] | Max. [3] |
|---|---|---|---|---|
| **Gross texture (%)** | | | | |
| Clay | 38.0 | 11.0 | 8.0 | 67.0 |
| Silt | 45.0 | 14.0 | 10.0 | 85.0 |
| Sand | 16.0 | 12.0 | 0.0 | 76.0 |
| **Chemical parameters** | | | | |
| pH (pore water) | 6.7 | 0.4 | 5.5 | 7.8 |
| pH (1:1) | 6.5 | 0.6 | 4.9 | 8.1 |
| Base unsaturation | 0.13 | 0.07 | 0.02 | 0.29 |
| Cation-exchange capacity—CEC (meq/kg) | 311.0 | 112.0 | 48.0 | 510.0 |
| Exchangeable acidity (meq/kg) | 34.0 | 15.0 | 13.0 | 105.0 |
| Exchangeable acidity/CEC (%) | 10.9 | 4.8 | 4.2 | 33.8 |
| **Gross biochemical composition (g/kg)** | | | | |
| Total carbon | 23.5 | 15.6 | 2.5 | 150 |
| Total nitrogen | 1.5 | 2.0 | 0.2 | 5.3 |
| Total sulfur | 4.4 | 3.8 | 0.3 | 20.6 |
| Total phosphorus | 0.799 | 0.143 | 0.460 | 0.134 |
| Dilute-acid-extractable phosphorus | 0.275 | 0.141 | 0.065 | 0.635 |
| Carbonates (CaCO3 equivalence) | 5.3 | 3.1 | 0.2 | 26.1 |
| **Minerals (g/kg)** | | | | |
| Ca, Calcium | 3.538 | 2.145 | 0.830 | 18.934 |
| K, Potassium | 1.376 | 0.287 | 0.211 | 2.596 |
| Na, Sodium | 10.343 | 4.832 | 0.955 | 42.472 |
| Mg, Magnesium | 3.492 | 1.523 | 0.411 | 9.541 |
| Fe, Iron | 0.651 | 0.234 | 0.045 | 2.555 |
| Al, Aluminum | 0.540 | 0.224 | 0.198 | 1.134 |
| Mn, Manganese | 0.135 | 0.115 | 0.010 | 0.779 |
| **Other minerals (mg/kg)** | | | | |
| B, Boron | 20.5 | 10.9 | 2.3 | 66.5 |
| Ba, Barium | 5.6 | 2.5 | 0.0 | 13.0 |
| Zn, Zinc | 9.8 | 3.3 | 0.0 | 23.5 |
| Cu, Copper | 6.2 | 3.0 | 0.0 | 32.8 |
| Co, Cobalt | 1.7 | 1.2 | 0.0 | 7.6 |
| Mb, Molybdenum | 1.2 | 0.6 | 0.0 | 2.6 |
| Pb, Lead | 3.9 | 2.3 | 0.0 | 12.4 |

[1] standard deviation; [2] minimum value; [3] maximum value. Values are means of combined subsamples from all ponds (n = 4). All measurements and percentages correspond to dry matter (DM).

**Table 2.** Correlation coefficients of bottom soil properties of seawater ponds at Garho Fish Farm.

|  | Calcium | Potassium | Phosphorus | Nitrogen | Carbon |
|---|---|---|---|---|---|
| Potassium | −0.24 |  |  |  |  |
| Total phosphorus | 0.16 | −0.23 |  |  |  |
| Total nitrogen | −0.80 ** | 0.18 | 0.10 |  |  |
| Total carbon | 0.43 * | 0.17 | −0.47 * | −0.49 * |  |
| pH | −0.09 | 0.14 | 0.05 | −0.10 | 0.63 |

* significant ($p < 0.05$), ** highly significant ($p < 0.01$).

**Table 3.** Yearly variability of physicochemical parameters of seawater ponds at Garho Fish Farm.

| Variable | Mean | SD [1] | Min. [2] | Max. [3] |
|---|---|---|---|---|
| Temperature (°C) | 31.6 | 4.6 | 23.6 | 36.9 |
| Salinity (ppt) | 35.8 | 1.1 | 34.2 | 38.9 |
| pH | 7.6 | 0.7 | 6.5 | 8.7 |
| Transparency (cm) | 32.5 | 5.8 | 28.0 | 45.0 |
| Dissolved oxygen, DO (mg/L) | 6.2 | 0.8 | 5.0 | 7.7 |
| Nitrates (mg/L) | 9.9 | 3.6 | 4.2 | 20.9 |
| Ammonia (µg/L) | 43.0 | 13.0 | 21.0 | 70.0 |
| Phosphates (µg/L) | 63.0 | 34.0 | 5.0 | 95.0 |
| Potassium (mg/L) | 39.2 | 4.4 | 21.1 | 48.5 |
| Calcium (mg/L) | 43.4 | 5.5 | 25.7 | 51.3 |
| Alkalinity (g/L) | 0.86 | 0.26 | 0.53 | 1.60 |
| Total hardness (g/L) | 19.0 | 0.68 | 17.89 | 25.81 |

[1] standard deviation; [2] minimum value; [3] maximum value. Values are the mean of all ponds (monthly measurements; n = 240).

**Table 4.** Correlation coefficients of water properties of seawater ponds at Garho Fish Farm.

|  | Phosphates | Nitrates | Potassium | DO | pH | Transpar. |
|---|---|---|---|---|---|---|
| Nitrates | −0.47 * |  |  |  |  |  |
| Potassium | −0.77 * | −0.24 |  |  |  |  |
| DO | 0.56 * | 0.86 ** | −0.14 |  |  |  |
| pH | −0.22 | −0.19 | −0.18 | 0.68 ** |  |  |
| Transparency | −0.77 * | 0.22 | −0.23 | −0.77 ** | −0.60 ** |  |
| Temperature | 0.46 * | 0.85 ** | 0.20 | −0.54 ** | 0.84 ** | 0.89 ** |

* significant ($p < 0.05$), ** highly significant ($p < 0.01$).

**Table 5.** Monthly variation of TSS (mg/L) recorded in seawater ponds at Garho Fish Farm.

|  | I | II | III | IV | V | VI | VII | VIII | IX | X | XI | XII |
|---|---|---|---|---|---|---|---|---|---|---|---|---|
| Mean | 114.3 [a] | 126.8 [a] | 109.9 [a] | 94.6 [a] | 106.5 [a] | 24.3 [b] | 110.8 [a] | 108.7 [a] | 105.5 [a] | 93.2 [a] | 36.3 [b] | 13.3 [b] |
| SD [1] | 14.4 | 23.8 | 28.6 | 49.3 | 12.1 | 14.4 | 47.2 | 19.6 | 39.0 | 3.5 | 38.2 | 5.3 |

[1] standard deviation; I–II—January–December; Different superscript letters indicate highly significant differences ($p < 0.01$).

**Table 6.** Monthly distribution and abundance of phytoplankton species/taxa recorded in seawater ponds at Garho Fish Farm.

|  | I | II | III | IV | V | VI | VII | VIII | IX | X | XI | XII |
|---|---|---|---|---|---|---|---|---|---|---|---|---|
| **Bacillariophyta** |  |  |  |  |  |  |  |  |  |  |  |  |
| *Bacillaria paxillifera* | + | ++ | + | + | + | + | + | + | + | + | ++ | + |
| *Biddulphia* sp. | − | − | − | + | + | − | − | − | + | − | − | + |
| *Chaetoceros atlanticus* | − | + | + | + | + | + | + | − | + | + | ++++ | − |
| *Chaetoceros decipens* | − | − | − | + | − | − | + | − | − | + | − | − |

**Table 6.** *Cont.*

| | I | II | III | IV | V | VI | VII | VIII | IX | X | XI | XII |
|---|---|---|---|---|---|---|---|---|---|---|---|---|
| *Chaetoceros teres* | − | − | + | − | + | − | − | − | − | − | ++ | − |
| *Climaconeis delicatula* | − | + | + | + | − | − | − | − | − | + | − | − |
| *Cocconeis* sp. | + | − | − | − | − | − | − | − | − | − | − | − |
| *Coscinodiscus thori* | + | − | − | − | − | − | + | + | − | − | − | − |
| *Cylindrotheca closterium* | − | − | + | − | + | − | − | − | − | − | + | − |
| *Ditylum brightwellii* | − | − | − | − | − | − | − | − | − | + | ++ | − |
| *Eucampia zodiacus* | − | − | − | + | + | + | + | ++ | + | ++ | ++ | − |
| *Gyrosigma spencerii* | + | ++ | ++ | ++ | ++ | + | − | + | + | + | ++ | + |
| *Navicula distans* | + | ++ | ++ | + | + | + | − | − | − | + | ++ | − |
| *Nitzschia acicularis* | + | ++ | − | + | + | + | − | + | + | + | + | − |
| *Nitzschia longissima* | − | + | − | − | − | − | + | + | − | − | − | − |
| *Odontella aurita* | + | − | − | − | − | − | − | − | − | − | − | − |
| *Pleurosigma normanii* | + | − | − | ++ | + | + | − | − | − | − | − | − |
| *Skeletonema costatum* | + | + | − | +++ | + | + | ++ | − | + | + | ++++ | − |
| *Sundstroemia setigera* | ++ | ++++ | ++++ | ++++ | +++ | +++ | +++ | + | ++ | ++++ | ++++ | + |
| *Thalassiosira nordenskioeldii* | + | + | − | − | − | − | − | − | − | − | ++ | +++ |
| **Dinoflagellata** | | | | | | | | | | | | |
| *Alexandrium* sp. | − | − | + | − | − | − | − | − | − | − | − | − |
| *Polykrikos* sp. | − | − | − | + | − | − | + | − | − | − | − | − |
| *Tripos furca* | − | − | − | − | − | − | − | − | − | + | ++ | + |
| **Prymnesiophyta** (*Coccolithus* sp.) | − | − | + | − | − | − | − | − | − | − | − | − |
| **Cyanophyta** | | | | | | | | | | | | |
| *Oscillatoria limosa* | + | + | + | ++++ | ++++ | +++ | ++++ | ++++ | ++ | ++++ | + | ++ |
| *Oscillatoria tenuis* | − | − | + | − | + | − | +++ | ++ | − | − | + | − |
| *Trichodesmium* sp. | − | ++ | + | − | + | − | + | + | + | + | − | − |

"++++" most abundant (71–100%), "+++" abundant (41–70%), "++" common (21–40%), "+" less common (1–20%), "−" absent (0%). Month-wise distribution represents the mean of all ponds; I–XII—January–December.

**Table 7.** Monthly distribution and abundance of zooplankton species/taxa recorded in seawater ponds at Garho Fish Farm.

| | I | II | III | IV | V | VI | VII | VIII | IX | X | XI | XII |
|---|---|---|---|---|---|---|---|---|---|---|---|---|
| **Copepoda** | | | | | | | | | | | | |
| Calanoids (*Acartia, Pseudocalanus*) | ++ | ++ | +++ | ++++ | +++ | + | ++++ | ++++ | ++++ | +++ | ++ | + |
| Cyclopoids (*Oithona*) | ++ | ++ | ++++ | ++ | +++ | + | +++ | ++ | ++++ | +++ | ++ | + |
| Harpacticoids (*Clytemestra*) | + | ++ | ++++ | ++ | + | + | ++ | + | ++ | + | + | + |
| Calanoid females | + | − | + | − | − | + | − | − | + | + | − | − |
| Cyclopoid females | + | − | − | + | − | − | − | − | + | + | − | − |
| Harpacticoid females | − | + | + | − | − | + | + | − | − | − | + | − |
| Copepod egg sacs | − | − | + | + | + | + | − | − | − | + | − | − |
| Copepod nauplii | + | ++++ | ++++ | ++++ | +++ | + | +++ | +++ | ++ | ++ | + | + |
| **Cladocera** (*Evadne* sp.) | + | + | + | + | + | + | + | + | − | − | − | − |
| **Amphipoda** (Hyperiids) | − | − | + | − | + | − | + | − | − | − | − | − |
| **Decapoda** | | | | | | | | | | | | |
| Penaeid nauplii | − | − | − | − | − | − | + | − | − | + | + | − |
| Penaeid mysis | + | + | + | − | − | − | − | − | + | − | − | − |
| *Acetes indicus* | − | + | + | − | − | − | − | − | − | − | − | − |
| *Lucifer* sp. adults | − | − | − | − | − | − | − | − | + | + | − | − |
| Euphausiid juveniles (krill) | + | − | − | − | − | − | − | − | + | − | − | − |
| Brachyura larvae (crabs) | − | − | − | − | − | − | − | + | − | + | + | − |
| Phyllosoma larvae (lobsters) | − | − | + | + | − | − | − | − | − | − | − | − |
| **Copelata** (*Oikopleura*) | − | + | − | − | − | − | − | + | − | + | + | + |
| **Doliolida** (*Doliolium denticulatum*) | ++ | ++ | + | + | ++ | + | + | + | + | +++ | + | + |
| **Salpida** (*Thalia democratia*) | − | − | − | + | − | − | − | + | − | − | − | − |
| **Pteropoda** | | | | | | | | | | | | |
| Creseis acicula | − | + | − | + | + | − | − | − | − | − | − | − |
| Limacina inflate | ++ | ++ | ++ | + | + | ++ | ++ | + | + | + | + | + |
| **Scyphozoa** (Medusae) | ++++ | + | + | + | + | + | + | − | + | + | ++ | + |

**Table 7.** *Cont.*

| | I | II | III | IV | V | VI | VII | VIII | IX | X | XI | XII |
|---|---|---|---|---|---|---|---|---|---|---|---|---|
| **Polychaeta** | | | | | | | | | | | | |
| Trochophore larvae | − | − | + | − | − | − | + | + | − | − | − | − |
| Siphonid larvae | − | − | + | − | − | − | − | − | − | − | − | − |
| Nereid larvae | − | − | + | − | − | − | − | − | − | − | − | − |
| **Nematoda** (worms) | + | + | + | + | + | + | + | + | + | + | + | + |
| **Rotifera** (Bdelloids) | − | − | − | + | − | − | − | − | + | + | − | − |
| **Tintinnida** | | | | | | | | | | | | |
| *Rhabdonella spiralis* | − | + | − | − | − | + | − | − | − | ++ | + | − |
| *Tintinnopsis* sp. | + | + | + | − | − | + | +++ | + | + | ++++ | ++++ | − |
| **Foraminifera** | | | | | | | | | | | | |
| *Globigerina* sp. | − | − | + | + | + | + | ++ | + | + | − | + | + |
| Other foraminiferans | ++ | + | + | ++ | + | + | + | ++ | +++ | ++++ | ++ | − |
| **Other unspecified taxa** | | | | | | | | | | | | |
| Fish larvae (hatched) | − | − | − | + | + | − | − | + | + | − | − | − |
| Eggs with embryo inside | − | − | + | − | − | − | − | − | − | + | − | − |
| Unidentified eggs | + | ++ | + | + | + | + | ++ | + | + | + | + | + |
| Unidentified worms | + | + | ++ | + | + | + | + | + | + | + | − | − |
| Unidentified larvae | + | + | − | − | − | − | + | − | − | − | − | − |

"++++" most abundant (71–100%), "+++" abundant (41–70%), "++" common (21–40%), "+" less common (1–20%), "−" absent (0%). Month-wise distribution represents the mean of all ponds; I–XII—January–December.

## 4. Discussion

The soil of studied ponds was characterized by predominantly good or moderate values of the most basic parameters (especially pH, CEC, or exchangeable acidity) recommended for aquaculture ponds [54]. The bottom soil texture consisted of high amounts of silt and clay, and a low sand content. Clay fractions, which include layered silicates and various hydrous oxides of aluminum, iron, and manganese, are found in highly weathered soils of coastal areas, where the weathering process of clay minerals results in the release of the alkaline earth metals, aluminum, iron, and silicate to the soil solution [55]. Silica is important for the development of diatoms, and most of the silica in ponds derives from the weathering of clay [56]. In addition, high clay content in soil can be linked with a high phosphorus fixation capacity, and it is often difficult to initiate planktic blooms in semi-intensive ponds with clay-heavy soils, even when intensive phosphate fertilization is used [57]. Nevertheless, it appears that the soil of the studied ponds had a generally desirable clay content [58,59].

In the studied fish ponds, soil carbonates may have occurred due to pond liming, although natural deposition of calcium and magnesium carbonates should not be excluded. However, it should be noted that the average recorded concentration of carbon (23.5 g kg$^{-1}$) was more than six-times lower than could be found in mangrove soils (up to 150 g kg$^{-1}$). Leftover feed, feces, and deceased plankton are the major source of organic carbon in aquaculture ponds [45]. These substances are classified as "labile organic matter" because of constant bacterial decomposition, which prevents their accumulation at high rates during the culture period [43]. Similarly, nitrogen is usually related to organic matter in the soil, as it was reported that 86% of nitrogen in seawater pond soil was organic nitrogen [56]. However, the present soil analyses have shown that total carbon was inversely correlated with nitrogen and phosphorus, and positively with calcium, while total nitrogen was negatively correlated with calcium. This is likely an indication that higher instances of total carbon in Garho Farm pond soil should be attributed to the presence of calcium carbonate, not organic carbon.

The presence of phosphorus is inevitable in pond soils. In our study, the total concentration of phosphorus was approximately three times higher than that of dilute-acid-extractable phosphorus. Two types of soil can generally be distinguished: acidic, where phosphorus precipitates as iron and aluminum phosphates, and alkaline, where phosphorus tends to precipitate as calcium phosphates [45]. It is presumed that the concentration of

dissolved phosphates might diminish in soil-water systems under aerobic conditions. The solubility of soil phosphorus tends to increase as a function of increasing concentrations of iron and aluminum phosphates [57]. Acids are particularly efficient in dissolving calcium phosphates, therefore the ratio of dilute-acid-extractable phosphorus to total phosphorus is lower in acidic soils than in alkaline soils. In addition, the concentration of calcium phosphates in soil is usually higher in brackish-water ponds than in freshwater ponds [60]. Nevertheless, proportion variations of different types of phosphates in pond soils may also remain in connection with farming activities, such as adding fish feed, fertilizers and other products.

Temperature is one of the overriding parameters of aquatic ecosystems, as it affects the chemical and biological processes therein, at all levels of the trophic chain [61]. The optimal range for plankton richness and rapid planktic biomass upsurge is considered to be 18–38 °C [62,63]. The water temperature of the studied ponds (23–37 °C) remained within this optimal range for planktic growth throughout the entire 12-month period (I–XII). Comparable yearly water temperature fluctuations were documented in India in Malgujari pond of Maharashtra (24–34 °C) [13] and in Puthukulam pond of Tamil Nadu (21–33 °C) [64]. Shorter studies have yielded similar results in Rajshahi, Bangladesh (18–30 °C, X–III) [65]; in Ifewara reservoir, Nigeria (24–31 °C; II-XII) [66]; and in Bohai Bay shrimp Ponds, China (15–33 °C, IV–IX) [16]. Naturally, seasonal influence is the major cause of water temperature fluctuations in such water bodies [51].

Water transparency strongly influences the penetration range of sunlight, which is crucial for primary producers; thus, the whole trophic chain is affected [51,67]. As shown in Table 4, transparency was highly correlated with temperature, possibly illustrating the effect of surging planktic communities on this parameter [68,69]. However, while the overabundance of phytoplankton itself may diminish water transparency in a feedback-like manner, it was shown that non-phytoplankton turbidity is the deciding factor in aquaculture ponds [70]. Accordingly, water in the studied earthen clay ponds remained highly turbid throughout the year (28–45 cm range), likely caused by run-off from pond dykes, but this is an issue which could be reduced, e.g., with rice-straw covering [71].

Other factors such as DO, pH, alkalinity, $CO_2$, and nutrients also affect plankton productivity in ponds [72,73]. The correlations of these parameters accurately mimicked the already well-known physicochemical dynamics of aquaculture ponds, such as the inverse relationship between DO and temperature or highly positive correlation between DO and pH [74]. Most importantly, the DO remained fairly stable throughout the year, never dropping below 5 mg/L (>75% saturation), therefore ensuring the survival of cultured fish. All in all, the measured parameters were all within the ranges deemed to be adequate for aquaculture ponds [54,75].

On a final note, we presume that the sometimes high variations of chemical parameters, such as phosphates (5–95 µg/L) or nitrates (4.2–20.9 µg/L), may be related to the fluctuations in the quality of the water flowing from the Garho channel. Apart from collecting the influx from nearby tidal creeks, the channel gathers household and agricultural wastewater from around the local area, which in general is a permanent practice in Pakistan, unfortunately [67,76–78]. Such domestic run-off intensifies during the monsoon season [79,80]. While phosphates and nitrates are crucial constituents of photosynthesis, such sewage often carries numerous pollutants (e.g., heavy metals), which is why there are both positive and negative aspects of the continuous water inflow adopted in the tested fish farm.

Among the identified phytoplankton, Bacillariophyta was consistently the most dominant group, followed by *Oscillatoria* spp. belonging to cyanobacteria. While diatoms are usually to be found in ponds during colder months, and their biomass is expected to increase under reduced light and temperature [81,82], cyanobacteria tend to grow faster at high temperatures and intense lighting [2,83]. It is known that Cyanophyta thrive in the summer because they produce UV-screening compounds which favor their growth at higher temperatures [84,85]. It appears that such was the case in the present study

because the relative abundance of Bacillariophyta species diminished in VII–IX, while *Oscillatoria* spp. peaked in VII–VIII, which coincided with the highest water temperature measurements. Similar observations were made in different studies focused on tropical ponds [65,86]. It should be mentioned, however, that the presence of ammonia may also be an important factor in the regulation of cyanobacteria in ponds [87], while alkalinity [65], phosphorus [2], and influx of other nutrients [88] are also very important for the proliferation of phytoplankton.

In contrast to diatoms, dinoflagellates seldom occurred in the studied ponds, as has also been observed in the coastal waters near Karachi [89,90], but this is definitely not a rule, as other studies on phytoplankton in Pakistani near-shore coastal waters [91] and tidal creeks [92] showed a high diversity and abundance of said dinoflagellates. It is known that this coastal belt receives high domestic industrial and agricultural discharges (originating from dumps into different river channels), which carry high quantities of dissolved nutrients, significantly contributing to a high primary productivity of these waters [92,93].

In terms of aquaculture ponds, the growth of Bacillariophyta may diminish after stocking due to higher pH and turbidity, and decreasing concentration of silicates in water, as opposed to the simultaneous growth of heterotrophic flflagellates, which graze on those diatoms [94,95]. On the other hand, diatoms may have a negative effect on the hatching rate of copepod eggs [96,97], but in coastal waters, this phenomenon was not confirmed at all [98]. Meanwhile, it was shown that the presence of some phytoplanktic communities may be beneficial for the biological performance and food assimilation efficiency of sparid larvae, as shown for *Mychonastes homosphaera* [99].

Copepods were clearly the dominant group of zooplankton throughout the study period, and were consistently found every month. Similarly to that revealed here, the dominance of calanoids and cyclopoids in ponds has been described before, but specifically only in high-altitude lakes [2,100]. Copepods are able to tolerate high radiation in the summer thanks to the elevated concentration of carotenoid pigments, mycosporines, and mycosporines-like amino acids [101,102]. Moreover, they are grasping feeders whose main source of nutrition is phytoplankton [51,103], although small copepods may have a preference towards protists [104,105].

The abundance and diversity of zooplankton is highly important for pelagic fish, especially during their larval and juvenile life stages, enabling the improvement of their growth and survival rates [90]. Eutrophication may even promote the expansion of omnivores at the expense of carnivores [106]. Considering that the studied ponds were examined for semi-intensive fish culture of regionally-important omnivorous species, the abundance of calanoids, cyclopoids, and harpacticoids appears to be of utmost importance. Generally, small-sized (<1 mm) copepods are a very important link in the marine food web and are perceived as excellent natural food for fish juveniles [104], being more adequate than the hatchery-mainstays *Artemia* sp. and rotifers, but also tougher to produce [95,107]. In the case of the *Acanthopagrus* spp. sparids, it has been specifically shown that juveniles feed mainly on this subclass of zooplankton, as studied on the black bream *Acanthopagrus butcheri* [108], blackhead seabream *Acanthopagrus schlegeli* [109], and also on the gilthead sea bream *Sparus aurata* [110,111]. In other words, copepods are nutritionally sufficient natural prey for culturing marine fish species [111]. Whereas, both planktic and benthic organisms are encountered in the gut of *Chanos chanos* [112,113], and this fish was therefore deemed adequate for polyculture ponds, especially in co-culture with shrimps [114,115]. Moreover, ciliates [116,117] and foraminiferans [118,119] may constitute an alternative source of nutrition for marine fish during the crucial, early developmental stages. Such increased abundance of these zooplanktic taxa in Garho ponds during the period of VII–XI could be beneficial for newly-hatched stocks of *Acanthopagrus* spp., as their spawning period commonly revolves around that specific timeframe [120,121].

The presence of rotifers was only sparsely registered throughout the year and occurred only in three of the monthly samplings (IV, IX–X). This should be regarded as favorable

information, as rotifer blooms are gross indicators of eutrophication of water bodies [122–124]. Even though such a simplistic approach has been challenged lately [125], especially with regard to unstable brackish water [126], this observation implies that the studied ponds remained free of that troublesome phenomenon over the course of the year. Meanwhile, the consistently low level of observed nematodes was also a positive sign, since many species are fish parasites [127,128], which in some rare cases may even be transmitted to humans [129].

The TSS calculation in this study should be treated only as a gross estimation of the overall planktonic abundance in the Garho ponds, given that there are numerous factors which influence the process of limnological change, especially in aquaculture fish ponds [130–133]. However, most noticeably, these basic results complied with the relative quantification of planktic species, which has shown the reduction of zooplankton richness in VI, and of both zoo- and phytoplankton in XII. We presume that temperature and light were the most likely causes for these events, with the zooplankton drop in VI resulting from excessive summer insolation, although small copepods have adopted many reproductive strategies to overcome their population losses [104], which may explain the rapid turnaround of their abundance which occurred in the following month (VII). Contrary to that, the lower insolation in winter resulted in the decline of phytoplankton at first, which successively led to the starvation of their zooplanktic grazers [134].

In terms of semi-intensive fish culture, this implies that the monitoring of planktic populations should be performed on a regular basis. Fish producers could then use such information with benefit to their production yields by modifying the amounts of artificial feed given to the fish, which would correlate with the amount of available natural food in the water. Additional efforts could also be made during the peaking winter to ensure the survival and sustain the growth of recently-hatched juveniles, for instance in the form of fertilizers to promote planktic production. However, such efforts should be done carefully, as they may cause a significant disturbance of water quality in the ponds [67,135,136].

Finally, future evaluations of the suitability of such ponds for semi-intensive aquaculture and the planktic communities within could also involve analyses of the turnover rate of nutrients within the trophic food web and the influx of nutrients into the produced species [137]. After all, the community structure of zooplankton may be affected not only by environmental conditions, but also simply by the presence of large numbers of planktivorous fish [138,139]. For instance, a lower abundance of *Daphnia* spp. and calanoids with concurrent high numbers of small Cladocera and cyclopoids was observed in lakes with a high planktivorous fish biomass [140]. Furthermore, larger invertebrates also predate on zooplankton and suppress the populations of smaller species, especially in the absence of fish [141]. In our study, however, no such patterns were observed, as the main copepod communities remained fairly abundant throughout the year and seemed to be rather affected by factors not related with predation. We presume that this might have occurred due to several reasons: (1) the stocking density of fish was too small to affect the planktic assemblages in a significant way; (2) the applied commercial diet satisfied the nutritional requirements of the fish, lowering their appetite for natural food; (3) the small, yet constant influx of seawater from the channel allowed for a perpetual replenishment of these planktic communities. Locally-executed studies in the near future should therefore focus on the assessment of the possible stocking capacity of such ponds, all the while minimizing the usage of commercial diets, in order to raise the cost efficiency of such semi-intensive polyculture by maximizing the natural productivity of these earthen water bodies.

## 5. Conclusions

The study has shown that clay-heavy earthen ponds in the coastal region of Sindh province show good promise for developing semi-intensive fish and shrimp aquaculture, which would certainly stimulate the expansion of the rural-based domestic economy of Pakistan. It is obvious that a great deal of effort is still needed in order to establish the most cost-efficient strategies for culturing commercially-important species. Nevertheless, the

present study is a first indication that the constant influx of nutrient-rich seawater flowing through channels from local tidal creeks provides a good environment for the sustainable growth of copepods, ciliates, and foraminiferans throughout the entire calendar year, even without the use of pond fertilizers. These zooplanktic subpopulations are preferable natural food sources for the vast majority of species showing the highest potential for mariculture production, such as the members of the Sparidae family. It should be mentioned, though, that these planktic populations need to be monitored carefully, because their abundance may periodically drop, as we have outlined in our observations for the year 2019. Thus, fish producers could quickly counter-react by modifying the amount of administered artificial feed, mitigating the temporal depletion of available natural food. Moreover, the use of fertilizers in agriculture is already a troublesome matter due to the resulting water and ground pollution; therefore, the possible avoidance of fertilizers for semi-intensive ponds definitely appears to be a solution worth consideration. Summing up, the studied ponds appear to be a highly adequate location for the development of coastal, semi-intensive fish production systems, including polyculture. Future initiatives in the Sindh region should therefore emphasize the promotion of possible benefits garnered from a growing aquaculture industry sector. Meanwhile, concurrent efforts should be focused on monitoring and counteracting the problems originating from uncontrolled discharge of urban and rural sewage, which could prove detrimental to the discussed coastal fish culture industry.

**Author Contributions:** Conceptualization, A.F. and G.A.; formal analysis, A.F.; investigation, A.F.; resources, G.A.; data curation, A.F. and G.A.; writing—original draft preparation, A.F.; writing—review and editing, R.K.; visualization, R.K.; supervision, G.A.; project administration, G.A.; funding acquisition, G.A. All authors have read and agreed to the published version of the manuscript.

**Funding:** This research was funded by Pakistan Agricultural Research Council, Islamabad, Pakistan, grant number AS 010-2021-23.

**Data Availability Statement:** Not applicable.

**Conflicts of Interest:** The authors declare no conflict of interest.

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
