# Peer review of "Assessment of Hydrobiological and Soil Characteristics of Non-Fertilized, Earthen Fish Ponds in Sindh (Pakistan), Supplied with Seawater from Tidal Creeks"

_water, doi:10.3390/w14132115_

Round 1

Reviewer 1 Report

The authors studied soil and seawater for mariculture activity.

1. It is suggest that the authors make the abstract more clear and summarize the manuscript well in conclusion section.

2. The quantification method is important in this manuscript. It should be described in detail to make the conclusion more clear.

Author Response

Firstly, we wish to thank the Reviewer for his/her valuable remarks and careful consideration of our manuscript.

In response to the issues raised by the Reviewer:

1. We have modified and expanded the Abstract (by over 60%) in order to clarify both the context and the conclusions of our study.
Likewise, we have implemented some adjustments in the Results and Conclusions sections, but these changes were not as drastic as in the Abstract. Specifically, we added one more conclusion which was previously missing and we also put more emphasis on the already existing conclusions. We hope that all of these changes meet the Reviewer's expectations.

2. According to the suggestion, we have enhanced the description of the planktic quantification method im the M&Ms section. We believe that now it will suffice.

Author Response

First of all, we wish to thank the Reviewer for this highly positive review of our paper. We are truly grateful for such opinion.

We have incorporated all the minor suggested linguistic corrections throughout the text, but we also did further changes according to the remarks specified in the other reviews.

In regard to the Reviewer's commentary for lines 307-308, we would like to reply that indeed, this was our speculation as we have no actual data to back this statement up. However, the latter sentences in this paragraph give more context to it (and contain some references), and we believe that the introduced changes within this sentence tone down the whole statement and turn it more into a prediction (we also changed the part at the beginning to "we presume").

Reviewer 3 Report

It is a very well written article and rich in interesting results. Studies that contemplate the variation of plantonic communities are very important for the development of IMTA systems, precisely due to the importance that these communities can have in the supplementary nutrition of cultured organisms, impacts on water quality and also as water quality parameters interfere in the dynamics and composition of these microbial communities. 
Personally, it would be very interesting to relate the results obtained with the performance results of farmed fish, but I understand that this is not the main objective of this study. 

Author Response

Dear Reviewer,

We are grateful for such a highly positive reception of our paper.
In response to Your last sentence: Yes, it would indeed be interesting to relate these results with fish performance results, but given the amount of information presented in this manuscript we have decided to publish it in a separate article. We hope that our decision will be met with Your acceptance.

Reviewer 4 Report

Overall, this paper is writing up in a professional scientific way. The authors have a sound knowledge of theoretical science. A case study is presented to study the Assessment of hydrobiological and soil characteristics of non-fertilized, earthen fish ponds in Sindh (Pakistan), supplied with seawater from tidal creeks. There is no review of the relevant literature (Literature Review) to highlight what approaches have already been employed in the study area. The authors should try to include this section. Conventionally used techniques are adapted. The methodology is addressed correctly.  References should be reduced because in their current form it shows a review report not a researcher article. It is suggested to remove irrelevant references from the paper.

Author Response

At first, we wish to thank the Reviewer for his/her insightful review. In return, we have addressed both of the Reviewer's concerns, both of which regarded references.

Firstly, in response to the apparent lack of a literature review about the study area, we have changed the last two paragraphs of the Introduction and have added 6 new references here, hoping that this better outlines the context of the study.

Secondly, we have addressed the issue of having too many references by reducing the number of cited papers throughout the whole manuscript, especially those that described local studies performed on natural water bodies - we deemed these studies irrelevant, thus, we deleted entire sentences from the Introduction and Discussion, all of which contained said information. As a result, the total number of references (even with the addition of the 6 new papers) dropped from the initial 191 to 141. We understand that this is still quite a lot, but we would have to remove a significant amount of papers which back up many of the statements and comments which we make throughout the Discussion. We hope for Your understanding on this matter.

Round 2

Reviewer 1 Report

The authors addressed some of the points raised by reviewer and this version looks good.

Reviewer 4 Report

I agree with the revised version.